# Advances in Genomics for Drug Development

**DOI:** 10.3390/genes11080942

**Published:** 2020-08-15

**Authors:** Roberto Spreafico, Leah B. Soriaga, Johannes Grosse, Herbert W. Virgin, Amalio Telenti

**Affiliations:** Vir Biotechnology, Inc., San Francisco, CA 94158, USA; Rspreafico@vir.bio (R.S.); Lsoriaga@vir.bio (L.B.S.); Jgrosse@vir.bio (J.G.); svirgin@vir.bio (H.W.V.)

**Keywords:** druggability, loss-of-function, CRISPR

## Abstract

Drug development (target identification, advancing drug leads to candidates for preclinical and clinical studies) can be facilitated by genetic and genomic knowledge. Here, we review the contribution of population genomics to target identification, the value of bulk and single cell gene expression analysis for understanding the biological relevance of a drug target, and genome-wide CRISPR editing for the prioritization of drug targets. In genomics, we discuss the different scope of genome-wide association studies using genotyping arrays, versus exome and whole genome sequencing. In transcriptomics, we discuss the information from drug perturbation and the selection of biomarkers. For CRISPR screens, we discuss target discovery, mechanism of action and the concept of gene to drug mapping. Harnessing genetic support increases the probability of drug developability and approval.

## 1. Introduction

For over 20 years, genomics has been used as a tool for accelerating drug development. Various conceptual approaches and techniques assist target identification, target prioritization and tractability, as well as the prediction of outcomes from pharmacological perturbations. These basic premises are now supported by a rapid expansion of population genomics initiatives (sequencing or genotyping of hundreds of thousands of individuals), in-depth understanding of disease and drug perturbation at the tissue and single-cell level as measured by transcriptome analysis, and by the capacity to screen for loss of function or activation of genes, genome-wide, using CRISPR technologies. In parallel to these areas of genomics/omics that we review here, proteomics and metabolomics are also influencing drug development, but are not addressed here.

The aim of this work is to present progress in the implementation of genomics in drug development. Any such effort of course represents a snapshot in time, as technologies are being brought to bear on the problem of diagnosis and treatment of human disease at an amazing rate. Old technologies may fade entirely if they become obsolete or may be retained for a specific use for which the technology remains well suited. As new technologies develop, they bring not only their unique contributions, but also provide opportunities for linking the new with the old to the benefit of both. In this complex data sciences space, it is of value to assess where things are at a specific point in time, the limitations of commonly used technologies, and how such technologies interact. In fact, these techniques do not compete with each other; rather, they are increasingly deployed and interpreted jointly. It should be underscored that data from genomic technologies are not a regular requirement for Investigational New Drug (IND) applications to regulatory agencies such as the US Food and Drug Administration [1]. They are, however, impacting the drug development program at many levels—we illustrate this concept in Table 1 by listing various queries that are now common in target and drug discovery.

Rather than reviewing the remarkable number of emerging genomic technologies, we take this opportunity to prioritize the discussion of more mature techniques with pointers to a selection of databases and resources. This review should be of interest to genomics and bioinformatic scientists that are interested in the field of drug development, and to pharmacologists and medical chemists looking to gain a better understanding of the implementation of large-scale genomics. Review of these more mature technologies provides an opportunity to identify challenges still unresolved.

## 2. Genome Sequencing and Genotyping

To better understand the potential for genome analysis in drug development, there is a need to spell out the properties of three techniques that are in current use: genome-wide association studies (GWAS) that use high-density genotyping of common variants (>1–5% of allele frequency in the population) and linkage analysis, exome sequencing capturing the coding sequences in ~1.5% of the human genome, and whole genome sequencing achieving good quality coverage of ~85% of the genome [2]. In contrast to genotyping arrays, exome and whole genome sequencing identify specific rare disease-associated variants (<<1% allele frequency) which may carry functional effects and be causal in disease. The technical specificities of the various technologies may determine the success in translating the variant discovery data into actionable information for drug development. The underlying concept [3] is to use the genome analysis to identify “experiments of nature”—naturally occurring mutations in humans that affect the activity of a particular protein target or targets—that can be used to estimate the probable efficacy and toxicity of a drug targeting such proteins, as well as to establish causal relationships between targets and outcomes.

### 2.1. GWAS and Drug Target Discovery

GWAS are credited for advancing the understanding of the biological basis of common disorders such as cardiovascular disease, diabetes, infectious diseases, inflammatory and autoimmune disorders. However, 80–90% of the phenotype-associated variants identified by GWAS are found within non-coding regions (e.g., intronic, ncRNAs, antisense, enhancer or insulator regions) [4] and are less likely to provide direct information on protein function. In addition, the contribution of single variants to a given phenotype is small, and in many cases, the biological effect is thought to be mediated by changes in expression. The variants profiled in SNP arrays can also be biased geographically or racially. Results based on these biased profiles may not be widely applicable and assumptions on drug effectiveness may not translate across all populations targeted for treatment. Despite the perceived limitations, GWAS data are broadly used across industry (Table 1).

Population studies of massive scale such as UK Biobank (https://www.ukbiobank.ac.uk/), provide phenotype-to-genotype association data across a wide range of phenotypes. Other resources, e.g., GWAS catalog (https://www.ebi.ac.uk/gwas/), list phenotype-specific associations. Large-scale studies increase the return of analyses via imputation (estimating missing genotypes to boost the power of detecting variants that are not genotypes with allele frequencies of 0.1–1%), reveal variants that change gene expression (e.g., expression quantitative trait loci, eQTLs), and expand the representation of human populations in the studies. More generally, GWAS data have been used to estimate the effect of genetic support for drug mechanisms on the probability of drug approval (see dedicated section below) [5]. However, the perceived limitations of GWAS for drug development are shifting attention towards sequencing (exome and genome) studies that capture the associations between rare variants and phenotypes, thereby providing a more direct evidence for a genetic target.

### 2.2. Exome, Gene Essentiality, and Drug Target Discovery

As discussed, exome analyses allow the identification of coding variants—rare and common—that can be assessed for the likelihood of functional impact (missense, loss of function) and for predicted deleteriousness via various predictive metrics. For example, one of the most commonly used predictive metrics is ‘Combined Annotation-Dependent Depletion’ or CADD, a score that ranks genetic variants on the basis of a wide range of data types [6]. The value of exome sequencing for diagnostics of rare disorders is well proven. The value for drug development is rapidly expanding on the basis of the following concepts: the identification of null (loss-of-function) variants and the notion of gene essentiality. A gene can be defined as essential when loss of its function compromises viability of the individual (for example, embryonic lethality) or results in profound loss of fitness [7]. Several computational methods are available to score gene essentiality—pLI (Probability of being Loss-of-function Intolerant) is commonly used to describe the tolerance of a given gene to the loss of function (LoF) on the basis of the number of protein truncating variants [8]. More recently, gnomAD shifted from using pLI to using the observed/expected score (o/e) for its ease of interpretation and continuity across the spectrum of selection. The concept of essentiality partitions the genome into roughly ~3000 genes that are thought to be essential for life or to maintain fitness, and ~3000 genes that may tolerate loss of function because they can be observed as null in apparently healthy adult individuals [7,9,10]. Of importance for drug development, the sequencing effort identifies individuals that have a favorable trait associated with gene loss (homozygous) or diminished (heterozygous/haploinsufficient) gene dosing. This translates to identification of drug targets for inhibition or antagonism. In a similar vein, sequencing may identify gene targets for agonists. This concept has achieved considerable success in the development of new lipid-lowering drugs guided by studies of the population genetics of *PCSK9* [11], *LPA* [12], *APOC3* [13,14], *NPC1L1* [15] and *ANGPTL3* [16,17]. In short, individuals with loss of function of these genes were protected from disease naturally while gain of function variants (*PCSK9*, *NPC1L1*, *LPA*) were associated with increased cardiovascular risk. The genetics of sclerosteosis, an autosomal recessive disorder characterized by bone overgrowth, also exemplifies the learnings from genetic observations; while homozygous individuals present pathologic increase in bone density, heterozygous carriers’ bone density is above the mean value of healthy age-matched individuals but is not pathological [18]. There are also examples of compelling genetics that have so far challenged drug development. Loss of function mutations of SCN9A (sodium voltage-gated channel α subunit) are associated with lack of pain perception, severe self-mutilation and often trauma-related death in teenage years. SCN9A gain of function (GoF) mutations cause severe pain syndromes: erythromelalgia, paroxysmal extreme pain, febrile seizures. However, although the genetic knowledge triggered intense drug discovery efforts over the last 15 years, they have not led to an approval so far—SCN9A inhibition is considered to be a “really hard problem of drug discovery” but still worthwhile. For several approved drugs genetic knowledge is supporting the indication (Table 2), even though this information often emerged only as the discovery efforts were already underway. Our own work on the genomics of obesity and on human metabolic gene variants underscores how sequencing campaigns can expand the catalogue of rare variants that have consequential effects on human disease phenotypes [19,20]. A parallel strategy for target discovery and drug development is applied in cancer [21,22], but it is out of the scope of the present review.

### 2.3. Whole Genome Sequence—Challenges in the Druggability of the Non-Coding Genome

Technical progress and a reduction of sequencing costs makes whole genome analysis increasingly attractive. On clinical grounds, whole genome sequencing is of particular interest for the study of rare genetic disorders that have no demonstrable finding after examining the coding regions [23]. Key elements in the non-coding genome include promoters, enhancers, insulators and determinants of chromatin structure and 3D conformation of the genome. So far, few diseases have been associated with rare deleterious variants in the non-coding genome [24,25]. These considerations notwithstanding, a high fraction of causative mutations in neurodevelopmental disorders such as intellectual disability and autism, belong to pathways of transcriptional regulation and chromatin remodeling [26]. This opens the debate on the druggability of the non-coding genome. There are new tools for the mapping of deleterious variants in the non-coding genome that could guide target selection much as it is the case for gain or loss of function variants in the coding genome [25]. Of particular interest is the targeting and use of non-coding RNAs (including miRNAs and lncRNAs) for the purpose of modulation of expression (reviewed in [27]).

## 3. Transcriptomics—Bulk and Single-Cell Sequencing

Transcriptional profiling of cells and tissues is perhaps the most common of all omics technologies. Its use in supporting drug development includes mapping responses to compounds, the interrogation of tissues and cells for the expression of a target of interest, and more recently, assisting with the identification of causal variants associated with clinical phenotypes. Moreover, transcriptomics has been explored as a source of biomarkers for stratification of patients in clinical trials.

### 3.1. Transcriptomics of Drug Perturbations

In the pharmaceutical industry, a prime application of transcriptomics is in extracting gene expression signatures upon treatment with drugs or other perturbations, often referred to as connectivity mapping [28]. Because traditional RNA-seq protocols are too expensive and laborious for the high-throughput nature of these efforts, cheaper and faster methods have been developed, such as the L1000 assay [29]—which experimentally measures the expression of nearly 1000 landmark genes by Luminex and computationally imputes unobserved transcripts—or genuinely transcriptome-wide chemistries such as PLATE-seq [30], DRUG-seq [31] and BRB-seq [32]. By leveraging multiplexing and 3′ counting, these optimized protocols allow screening hundreds of perturbations in relevant cell types and in a time-course setting. Despite this increased throughput, screening campaigns still need to be performed in discrete batches. Batch effects (and, more generally, technical factors affecting reproducibility) remain the major analytical hurdle in extracting insight from any drug-profiling dataset, either based on transcriptomics [33,34] or other readouts (WEB: https://www.kaggle.com/c/recursion-cellular-image-classification). Proprietary drug profiling endeavors can be modeled after large public efforts such as LINCS Connectivity Map 2.0, which, in addition to releasing openly accessible datasets, sets best practices and provides analytical tools (WEB: https://clue.io/). Connectivity maps can be used to cluster drugs by transcriptional outcomes or to find drugs either mimicking or reverting transcriptional phenotypes of interest, such as those resulting from disease, thereby facilitating drug repositioning [35], as was the case of a novel suggested indication for celastrol in the treatment of obesity [36]. As the field evolves, the dimensionality of datasets will continue to grow; a recent report leveraging nuclear hashing coupled with single-cell combinatorial indexing enabled drug profiling at single-cell level resolution. This approach, called sci-Plex, promises to interrogate the heterogeneity of transcriptional responses to compounds at massive scales [37].

### 3.2. Bulk and Single-Cell RNA Sequencing to Characterize Drug Targets

Transcriptomics can offer insights into mechanisms of action and off-target effects. Compared to other -omics technologies constrained by cell types or numbers, RNA sequencing does not significantly limit experimental designs, thereby allowing the selection of the most physiological in vivo and in vitro models. Such flexibility stems from a diverse array of protocols ranging across low inputs [38], bulk or single-cell interrogation [39] and, more recently, even spatial transcriptomics [40]. In the era of arrays or bulk RNA-seq, getting insight from tissue transcriptomics was impaired by cell type heterogeneity. This issue prompted the development of computational methods to deconvolve aggregate tissue transcriptomes into constituent cell type-specific profiles [41]. However, reference-free, full deconvolution yielding both per-cell type and per-sample signatures as recently achieved for “digital” DNA methylation [42] remains elusive for “analog” transcriptomics, with one of the most accurate methods still yielding per-group (rather than per-sample) cell type signatures [43]. These concerns will be fading as single-cell RNA-seq reaches maturity, scale and affordability, allowing direct measurement of complex tissues at cell resolution from numerous samples. Protocols based on microwells [44] or split-pool barcoding [45] are particularly promising in this regard. The recent publication of high-quality single-cell atlases from several organs, coalesced by the Human Cell Atlas initiative (https://www.humancellatlas.org/), demonstrates the feasibility of deploying single-cell RNA-seq in large projects, and will ultimately result in a shared reference map of the entire human body [46], which can be used to query the expression patterns of targets of interest. When transcriptomics is used to shed light on complex biological processes, it is important to note that diverse pathways might converge towards similar transcriptional states. Therefore, it is often impossible to decipher the unknown insult that resulted in the measured transcriptome profile from a single experiment. In fact, substantial experimental triangulation and perturbation of the system are needed to achieve this goal [47,48]. In practice, this translates to complex experimental design and robust computational frameworks to decipher the effect of individual perturbations and the marginal contributions of genetic interactions on the level of each transcript, program, and cell state [48].

### 3.3. Biomarkers from Transcriptome Data

Another popular use case for transcriptomics relates to identifying biomarkers for cohort stratification or prediction of therapeutic outcomes, which is the foundation of the personalized medicine paradigm. To this aim, samples from patients (typically PBMCs or biopsies) are profiled by RNA-seq, and the gene-sample expression matrix is fed to supervised machine learning algorithms for classification and regression [49]. The major plague affecting these endeavors is limited statistical power, as the literature is dominated by a constellation of small to medium-sized studies as opposed to fewer but adequately powered studies. This contrasts with genetic association studies, where rigid statistics and a more developed field have led to universally accepted best practices. Hurdles due to limited sample size might be mitigated by computational approaches that reduce the dataset dimensionality and identify major trends (“gene modules”), such as WGCNA [50]. Alternatively, the availability of numerous but individually underpowered transcriptomics studies is naturally conducive to meta-analysis as the prime analytical tool to select biomarkers [51].

### 3.4. Linking Transcriptome to Genome Data

Mohammadi et al. [52] used transcriptome data in association with genome data to facilitate the identification of genes that are profoundly dysregulated and associated with disease. The approach leverages the Genotype-Tissues Expression (GTEx) population data to identify causal genes. While the original application of this technology is in pipelines that use RNA-seq data for the diagnosis of rare diseases, the conceptual approach can be extended to identifying novel genotype–phenotype relationships leading to the identification of new drug targets.

## 4. CRISPR-Based Technologies

Whereas genome-wide association studies rely on the distribution of naturally occurring variants to link human genes or genomic loci to a particular phenotype or function, CRISPR-based genome editing makes it easy to create targeted genetic perturbations at scale and screen for a phenotype of interest. Beyond its wild-type effect of disrupting specific genetic loci by DNA cleavage (CRISPRko), first demonstrated in 2012, RNA-programmable genome-targeting by CRISPR/Cas9 has been harnessed to inhibit or activate transcription (CRISPRi/CRISPRa), edit specific nucleotides, and modify epigenetic states [53,54]. Despite the variety of available genome-wide libraries for CRISPR-based genetic perturbations (http://www.addgene.org/crispr/libraries/), screening for targets relevant to disease or drug mechanism-of-action are largely limited by the suitability and scalability of available model systems [53,55,56]. These limitations aside, CRISPR screens have quickly driven target prioritization in a variety of disease models and clarified the targets, enhancers, and resistance genes for existing drugs [57].

### 4.1. Genome-Wide CRISPR Screens for Drug-Target Discovery

The development of pooled screening approaches for genome-scale RNA interference-based loss-of-function screens paved the way for the rapid adoption of genome-wide CRISPR screens for drug-target discovery [58]. Pooled screening enabled the simultaneous profiling of a genome-wide library of sequence-specific perturbations in a single experiment and leveraged massively-parallel sequencing to deconvolute which perturbations were associated with the phenotype of interest. There are limited studies benchmarking widely-used methods for scoring gene-level hits from genome-wide CRISPR screens and optimal methods may vary depending on the screen design and type of perturbation—for example, gene knockout versus transcription activation or inhibition [59,60,61,62]. The quality of genome-wide libraries, largely dependent on algorithms for optimal sgRNA selection, has also been shown to impact screen performance in benchmarking based on recovery of essential genes in negative selection screens [63]. Although the tools for the library design and analysis of CRISPR screens continue to evolve, large-scale projects which previously leveraged RNAi for genome-wide loss-of-function screens have largely converted to CRISPR-based screens due to the significant gains in on-target specificity [64], and the advantages of full knockout versus hypomorphs. A key example is the Cancer Dependency Map Project which aimed to identify therapeutic targets by systematic identification and comparison of essential genes across hundreds of cancer cell lines [65].

To date, the majority of integrative analyses efforts and open-source databases of CRISPR screen data such as Project Score (https://score.depmap.sanger.ac.uk/) and DepMap (https://depmap.org/portal/) have been applied to cancer drug discovery [66,67]. Because cancer cell lines can be readily expanded to achieve sufficient representation (>500X) of cells targeted by a specific perturbation, they are more readily used as models for primary genome-wide CRISPR screens (https://orcs.thebiogrid.org/, [68,69]). In addition to cancer, CRISPR screens have driven target prioritization for diseases as diverse as Alzheimer’s disease [70], Huntington’s disease [71], Type II diabetes [72], mitochondrial disorders [73], and ciliopathies [74]. Of note, many candidates for host-directed therapy have been identified based on independent genome-wide CRISPR screens for the host-dependency or restriction factors of a diverse array of clinically-relevant pathogens [75], such as HCMV [76], DENV [77,78]), Enteroviruses [79], IAV [80,81], HBV [82], HIV [83], Norovirus [84,85], SARS-CoV-2, WNV [86,87]), Zika [78,88], Legionella [89], Salmonella [90], and Mycobacteria. Application of CRISPR screens for target discovery is primarily bottlenecked by the optimization of relevant assay models. In particular, screens in primary cell and in-vivo models, which are limited by cell divisions and cell numbers, remain technically challenging and are typically restricted to more focused libraries targeting druggable gene families (kinases, GPCRs, ion channels) or primary screen hits from genome-wide in-vitro screens in cell line models [91,92,93]. Despite these hurdles, notable immuno-oncology targets have been discovered by in-vivo CRISPR screening in mouse xenograft models for modulators of cancer immunotherapy [94,95,96].

### 4.2. Gene-to-Drug Mechanism-of-Action

Functional genomics screening in yeast established the paradigm which links small molecule or drug sensitivity to the expression level (knockout, inhibition, or activation) of its target(s) [97]. Thus, orthogonal validation of CRISPR-based genetic perturbations by chemical perturbations of the corresponding protein target or pathway, using existing drugs or chemical probes, has expedited target triage [98,99]. Open-source and commercial resources such as OpenTargets [100], DGIdb [101], ChEMBL [102], GuideToPharmacology [103], Drugbank [104], Clarivate Integrity, GVK Excelra GoStar, and Citeline Pharmaprojects, which map clinical-stage drugs and active compounds to target human proteins, facilitate gene-to-drug validation workflows as well as repurposing of existing compounds for alternative indications [105,106,107]. Combining CRISPR screens with drug or compound treatment has also been used to validate on-target specificity [108,109] and to clarify the mechanism of action for poorly characterized drugs [97]. For example, combined CRISPRi/a chemical-genetic screening resolved microtubule destabilization as the mechanism of action for rigosertib, a phase 3 drug for the treatment of myelodysplastic syndrome [110].

### 4.3. CRISPR Screens and Drug Response

Beyond target prioritization and validation, CRISPR screens combined with drug treatment can reveal genes which enhance or suppress treatment effects [99,111]. Genes that confer resistance represent targets for synergy. For example, kinome-wide CRISPR screens led to identification of ILK inhibition as an enhancer of FGFR inhibitor response in gastric cancer [112]. Similarly, CRISPR screens focused on epigenetic modifiers led to the discovery that inhibition of Asf1a, a histone chaperone, sensitizes lung adenocarcinoma tumors to anti-PD1 treatment [113]. CRISPR screens for chemogenetic interactions are no longer limited to gene-level associations. The development of CRISPR base-editing screens are poised to discover human genetic variants of therapeutic relevance and advance pharmacogenomic annotation efforts [114,115]. Proof-of-concept pooled screening of 52,034 clinically-observed variants in 3854 genes in the context of cisplatin treatment resulted in the expected identification of loss-of-function variants in DNA repair genes (BioRxiv: https://doi.org/10.1101/2020.05.17.100818). More generally, CRISPR-based deep scanning mutagenesis and population genomics are converging on the goal of generating and interpreting variation of unknown significance in genes of medical relevance [116].

## 5. Genetic Support and the Probability of Drug Approval

There have been a number of publications that assess whether receiving genetic support was influential for the process of drug approval and for drug efficacy. DrugBank (https://www.drugbank.ca/, accessed 26 June 2020) indicates that there are 2631 approved small molecule drugs associated with 2611 unique targets. There are also 2162 approved biologicals that associate with 319 unique targets. The robustness of the association of a given drug to a genetic target is critical for estimating the contribution of genetic information to druggability; there is always the implicit limitation that drugs may interact with more than one target. Nelson et al. [117] concluded, on the basis of historical pipeline data from the Informa Pharmaprojects database, that drugs developed with knowledge of direct genetic evidence (see below) were twice as likely to result in approval. More recently, King et al. [5] used GWAS association data, OMIM gene-trait links and a formal statistical framework to give further support to the observation. Specifically, King et al. found that when causal genes are clear (Mendelian traits and GWAS associations mapped to coding variants), approval rates doubled. In these studies, genetic evidence of association between gene and target was defined by the similarity of the clinical trait and the drug indication as measured by semantic similarity in the MeSH vocabulary. Overall, these works indicated that investment into genomics for the purpose of improving the fraction of successful drug targets appeared to be well founded [5].

A recent analysis by gnomAD [118] gave a different and more nuanced report on the association of genetic evidence and druggability. Here, the analysis centered around the value of knowing the tolerance to mutation or essentiality of a gene for predicting the druggability of a target [10]. The hypothesis is that most essential, constrained/conserved genes would be poor targets because of adverse consequences of agonism/antagonism on toxicity. For this analysis, using DrugBank, they narrowed the number of targets that can be defined as having a top-ranked mechanistic target for approved drugs to 386 [118]. They concluded that targets of approved drugs range from highly constrained (~essential) to completely unconstrained and that a highly deleterious knockout phenotype is compatible with a gene being a drug target [118]. On this basis, there is no guidance to the use of essentiality metrics for decisions on potential drug targets.

Population genomic data can also be used to characterize prioritization of drug target sites in the context of protein structures [119]. We have previously analyzed the 3D intolerance to mutation of 97 proteins that included known drug targets with a bound ligand and proteins with known allosteric sites [120]. Active sites were most constrained, followed by allosteric, protein–protein interaction, and ligand-binding pockets. There was unequal distribution of mutation-tolerant and intolerant binding sites across therapeutic classes. For example, antineoplastic and immunomodulating agents preferentially target mutation-intolerant sites. We speculated that the identification of mutation-intolerant 3D sites and domains in drug targets could be exploited for rational drug design and for analysis of drug screening results [120].

## 6. Conclusions and Future Prospects

In 2013, Plenge et al. [3] listed criteria that underlie the principles of gene–drug pairing for drug development. These include the unequivocal association of a gene with the medical trait of interest and, in turn, the correspondence of the genetic trait to the clinical indication for a drug. Complementary criteria include the more traditional attention to the druggability of the gene target. There is broad consensus on the value of genetic information, but there are also a number of challenges (Box 1). A particular consideration is the rapid increase in data and the need for effective tools to integrate various data modalities and sources of knowledge. Drug development includes today various data science approaches (network biology, machine learning and deep learning) that leverage the large volumes of data generated by the different genomics technologies [121,122,123]. Although this review does not discuss the impact of genomics in later stages of drug development (i.e., clinical trials), many of the tools considered in the present review are valid for patient stratification and pharmacogenetics. Genomics (omics) technologies are becoming an integral component of drug development. They respond to the goal of compressing drug development timelines, and reflect the attention to personalized care.

Box 1Benefits and challenges of genetics- and genomics-based drug development. Modified from https://www.amgenscience.com/items/genetics-driven-research-benefits-and-challenges/.
**Benefits**
More relevant to human biology than animal models of disease.Insights into safety and potential side effects.Possible higher approval and clinical success rates.Increased potential for first-in-class therapies.Facilitated target validation.

**Challenges**
Targets may involve unexplored biology.Targets may be difficult to drug—no precedent.For rare genetic variants, long-term health consequences may be unknown.Though non-essential genes are intuitively more attractive for development, there are successful drugs acting on genes that do not tolerate genetic variation.Need to improve on data integration and algorithms for better predictive models.


## Figures and Tables

**Table 1 genes-11-00942-t001:** Genomic data impacting target identification and drug development. Common uses of genomic, transcriptomic and CRISPR editing data in industry. This table describes selected queries and representative sources based on the mature techniques described in this review.

Query	Representative Sources	Expected Output	Implication for Drug Development
Relevant population data for a given target	UK biobank (https://www.ukbiobank.ac.uk/), GWAS catalog (https://www.ebi.ac.uk/gwas/)	Genetic evidence of association between gene and target (similarity between the clinical trait and the drug indication)	Target identification, druggability
Genetic diseases	OMIM (https://omim.org/)	Evidence for severe consequences of genetic variants	Druggability, consequences on long-term drug action and safety
Null individuals	gnomAD (https://gnomad.broadinstitute.org/)	Identification of individuals in the general population that tolerate heterozygous or homozygous loss of function	Druggability, consequences of long-term drug action and safety
Relevant tissue expression	GTEx (https://www.gtexportal.org/home/)	Target is pertinent to the disease tissue	Target identification, validation
Relevant cell expression	Human cell atlas (https://www.humancellatlas.org/)	Target is pertinent to the cell implicated in pathogenesis	Target identification, validation
Expression perturbation	LINCS (http://www.lincsproject.org/)	The target responds to relevant perturbation(s)	Target identification, validation, mechanism of action
Target relevance and triage	CRISPR KO (https://depmap.org/portal/depmap/)	The target is relevant to in vitro or in vivo experimental endpoints	Target identification, validation
Gene-to-drug matching and precedent	Open Targets Platform (https://www.targetvalidation.org/)	The target genetic perturbation matches the putative drug perturbation endpoints	Druggability, repurposing, chemical matter

**Table 2 genes-11-00942-t002:** Selected examples of genetic conditions supporting the indication of approved drugs. Additional historical gene–drug pairs can be found in Plenge et al. [3]. GoF: gain of function; LoF: loss of function. CHD: coronary heart disease. eQTL: expression quantitative trait locus.

Gene (Protein)	Genetic Defect/Variant	Human Phenotype	Drug: Indication	Mechanism of Action
*PCSK9;* proprotein convertase subtilisin/kexin type 9	GoF (deleterious), LoF (protective)	GoF: familial hypercholesterolemia and CHD. LoF: lower LDL-C and CHD incidence	Evolocumab (Amgen) and Alirocumab (Regeneron): Familial hypercholesterolemia	PCSK9 cleaves the hepatic LDL receptor in the endosome depending on cellular cholesterol levels. PCSK9 inhibition leads to increased LDL receptors and hence clearance of LDL particles from the circulation
*NPC1L1;* Niemann-Pick C1-Like 1	GoF (deleterious), LoF (protective)	Heterozygote carriers of LoF alleles have a very modest reduction in LDL cholesterol but a large reduction of cardiovascular risk	Ezetimibe (Merck): Hypercholesterolemia	Ezetimibe inhibits the intestinal absorption of cholesterol from the diet and from the bile. In addition, it reduces the uptake of plant sterols. Shifting the ratio between cholesterol uptake and de novo synthesis might be a factor explaining the discrepancy between the moderate effect on LDL-cholesterol and the cardiovascular benefits.
*ANGPTL3;* angiopoietin-like protein 3	LoF (protective)	Familial combined hypolipidemia: reduced blood lipids, including LDL, VLDL and HDL cholesterol and triglycerides resulting in significantly lower risk of coronary artery disease	Evinacumab (Regeneron): Familial hypercholesterolemia	Neutralization of ANGPTL3 which is an inhibitor of lipoprotein lipase and endothelial lipase. In addition, it activates integrin αVβ3 which contributes to intima proliferation.
*LPA;* Lipoprotein(a)	GoF (deleterious), LoF (protective)	High plasma concentrations of Lp(a) as well as genetic variants which are associated with high Lp(a) concentrations are both associated with cardiovascular disease which very strongly supports causality between Lp(a) concentrations and myocardial infarction, stroke, peripheral vascular disease and childhood thromboembolism	AKCEA-APO(a)-LRx (Ionis) is an antisense drug that inhibits the production of apolipoprotein(a), thereby reducing Lp(a).	Reduction of hepatic Lp(a) translation and secretion resulting in reduced circulating levels and consequently in reduced cardiovascular risk.
*LEPR;* Leptin receptor	LoF (deleterious)	Severe early-onset obesity, major hyperphagia, hypogonadotropic hypogonadism and neuroendocrine/metabolic dysfunction	Metreleptin (Aegerion), a leptin analogue, and REGN4461 (Regeneron), a leptin receptor agonist for lipodystrophy and obesity.	REGN4461 is a fully human monoclonal antibody that is an agonist to the human leptin receptor (LEPR). In lipodystrophies the adipokine leptin is not adequately produced leading to severe hyperlipidemia and insulin resistance with consequential diabetes which is very difficult to manage
*MC4R;* Melanocortin 4 receptor	LoF (deleterious)	Early onset obesity due to increased appetite and reduced energy expenditure; increased body height.	Setmelanotide (Rythym): pro-opiomelanocortin (POMC) deficiency obesity and leptin receptor (LEPR) deficiency obesity	Setmelanotide is a peptide agonist of MC4R, a GPCR in the hypothalamus mediating satiety. In addition, activation of MC4R enhances sympathetic tone, metabolic rate and blood pressure, an obstacle for previous MC4R agonists. Setmelanotide does not elevate blood pressure or heart rate.
*PPARG;* peroxisome proliferator activated receptor γ	LoF (deleterious)	Familial partial lipodystrophy 3: partial lipodystrophy affecting extremities. increased adiposity on body and intraperitoneally, acanthosis nigricans, insulin resistance with dyslipidemia	Thiazolidinediones (Rosiglitazone, Pioglitazone): Diabetes type 2	Differentiation of adipocytes leading to increased insulin sensitivity, glucose uptake and secretion of adipokines (leptin, adiponectin).
*SOST;* Sclerostin	LoF (homozygous:disease, heterozygous: protective)	Sclerosteosis is characterized by bone overgrowth with high bone mineral density. It can lead to facial distortion, syndactyly and elevated intracranial pressure with sudden brain incarceration and death	Romosozumab (Amgen): Postmenopausal osteoporosis.	Sclerostin is a negative signal secreted from osteocytes acting as an antagonist on LRP5/6 receptors on osteoblasts negatively regulating Wnt-mediated differentiation and activation of osteoblasts. Neutralization of sclerostin leads to increased osteoblast activity and bone formation.
*SLC22A12;* Urate transporter 1	LoF (deleterious)	GoF: Uric acid elevated (hyperuricemia) leading to gout. LoF: Hyperuricosuria and nephrolithiasis	Lesinurad (Ironwood): Hyperuricemia	Inhibits reabsorption of uric acid in the proximal tubule of the nephron with elevated urate excretion
*XDH;* Xanthine oxidase	LoF	Xanthinuria	Allopurinol: Gout	Blockade of the oxidations hypoxanthine → xanthine → uric acid results in reduced urate production and increased urinary xanthine excretion.
*IL4; IL13; IL4Ra;* Interleukin-4, -6 and IL4 receptor α	eQTL (all 3 genes) and GoF (IL13 and IL4Ra)	Airway obstruction in asthma patients, asthma severity. IgE elevation	Dupilumab (Regeneron): Asthma, atopic dermatitis, chronic rhinosinusitis with nasal polyposis	Dupilumab blocks binding of IL-4 and IL-13 to IL-4α receptor which is used by both ligands. Previous attempts to neutralize IL-4 signaling only were not efficacious.
*NLRP3,* NOD-, LRR- and pyrin domain-containing protein 3	GoF (deleterious)	Cryopyrin-associated periodic syndrome (CAPS) is an autoinflammatory disorder characterized by systemic, cutaneous, musculoskeletal, and central nervous system inflammation	Canakinumab (Novartis); Anakinra (Amgen); Rilonacept (Regeneron): Rare and serious auto-inflammatory diseases in adults and pediatric patients	AB, endogenous receptor antagonist and decoy receptor neutralizing IL-1β, which is, together with IL-18, the product of the activated NLRP3 inflammasome. Canakinumab was shown to reduce cardiovascular events in a secondary prophylaxis study, to slightly increase sepsis occurrence, and unexpectedly to reduce several cancer diagnoses including lung cancer.
*F10,* Factor X	LoF (deleterious)	Hemophilia with variable penetrance. Prolonged activated partial thromboplastin time and prothrombin time	Rivaroxaban (Janssen), Apixaban (BMS): Anticoagulation as secondary prevention of stroke and myocardial infarct. Andexanet Alfa (Portola): antidote for FXa inhibitors	Blocking binding pockets S1/4 required for binding and cleavage of FXa’s substrate prothrombine. Andexanet is a proteolytically inactive recombinant FXa acting as a decoy receptor for the small molecule inhibitors.
*CFTR;* cystic fibrosis transmembrane conductance regulator	Missense, LoF (deleterious)	Cystic Fibrosis	Tezacaftor, Elexacaftor, Ivacaftor, Lumacaftor as fixed combinations (Vertex): Cystic fibrosis	Ivacaftor: gate opener (potentiator); Lumacaftor, Elexacaftor and Tezacaftor: chaperone and trafficking (corrector)
*HCRTR2;* Hypocretin receptor 2	LoF (deleterious) in dog breeds. LoF mutations have been detected in the ligand, HCRT.	Narcolepsy (sudden loss of wakefulness, daytime sleepiness, disturbed sleep patterns mainly due to autoimmune reactions against orexin secreting neurons	Lemborexant (Eisai), Suvorexant (Merck): Insomnia due to difficulties with sleep onset or maintenance	Dual antagonism of HCRTR1 and 2 receptors block the wakefulness signal mediated by the neuropeptides hypocretin 1/2 (also known as orexin A/B) temporarily for sleep induction and maintenance.
*SGLT2;* Sodium glucose cotransporter 2	Missense, LoF (protective)	Familial renal glucosuria	Dapagliflozin (AstraZeneca); empagliflozin (Boeringer/Lilly), canagliflozin (Mitsubishi/J&J): Type 2 diabetes; heart failure with reduced ejection fraction.	Inhibition of SGLT2 abrogates the glucose reabsorption from the primary filtrate in the proximal tubule. As a result, glucose is excreted with the urine. Remarkably, SGLT2 inhibitors are the only anti-diabetic drugs with clearly demonstrated cardiovascular benefits.
*JAK1;* Janus kinase 1	LoF (deleterious)	Deletion of Jak1 is perinatally lethal in mice. A single patient with homozygous missense mutations in the pseudokinase domain established its role for the recruitment of JAK2 which is essential for IFN-γ signaling. This patient suffered from combined immune deficiency with atypical mycobacterial osteomyelitis, sinopulmonary and skin infections, flat warts, and scabies.	Tofacitinib (JAK1/3, Pfizer)), Baricitinib (JAK1/2, Eli Lilly), Upadacitinib (JAK1, AbbVie): Rheumatoid arthritis.	JAK1 is involved in signal transduction of IL-2, IL-4, IL-7, IL-9, IL-15, IL-21, IL-27; IL-6 and IL-10 families as well as type I and II interferon. Two members of the JAK family work in common for specific signal transduction cascades: JAK1/3: IL-2, IL-4, IL-15, IL-21; JAK1/2: IL-6, IFN-γ; JAK1/TYK2: IL-10, IFN-α; JAK2/2: IL-3, GM-CSF; JAK2/TYK2: G-CSF
*HCN4;* Hyperpolarization-activated cyclic nucleotide-gated channel 4	LoF (deleterious) GoF (deleterious)	Expression in sinu-atrial, atrio-ventricular node and Purkinje fibers explains the various cardiac phenotypes affecting conductance and pace-making	Ivabradine (Amgen): Chronic heart failure.	Ivabradine is a non-selective blocker of HCN1/2/3/4 cation channels. The label of “a selective bradycardic agent” refers to the absence of effects on other hemodynamic parameters. Very limited crossing of the blood–brain barrier avoids effects on the CNS thus providing some selectivity for the heart.

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
