# Peer review of "Advances in Genomics for Drug Development"

_genes, 2020, doi:10.3390/genes11080942_

Round 1
Reviewer 1 Report
As a reviewer, I enjoyed reading this Review article. It is well-written logically and in a coherent manner, and the authors summarized well the advances in genomics for drug development.
I can suggest just a few points to improve this paper.
Minor comments;
- Line 38 in the Introduction section, please correct the repetition of “emerging”.
- Line 325, a bit difficult to understand the formula.
Author Response
Many thanks for the positive review.
Here are the answers to the minor queries:
Line 38 in the Introduction section, please correct the repetition of “emerging”.
>>>Corrected
Line 325, a bit difficult to understand the formula.
>>>opted to remove the formula from the text
Reviewer 2 Report
Advances in genomics for drug development
Comments:
This manuscript describes the contribution of population genomics to target identification, the value of bulk and single cell gene expression analysis for understanding the biological relevance of a drug target, and genome-wide CRISPR editing for the prioritization of drug targets.
However, I have a few issues that I would like to raise with the authors.
- The abstract is too small may add some more words and explain the genomics for drug development described.
- Introduction part is very conceptual and informative. However, this review has some limitations.
- The authors may add future prospective and challenges.
- Check syntax errors and misspells in whole manuscript.
The overall manuscript is good in shape but before going to publication check spells.
Author Response
- The abstract is too small may add some more words and explain the genomics for drug development described.
>>> Added text:
In genomics, we discuss the different scope of genome-wide association studies using genotyping arrays, versus exome and whole genome sequencing. In transcriptomics, we discuss the information from drug perturbation and the selection of biomarkers. For CRISPR screens, we discuss target discovery, mechanism of action and the concept of gene to drug mapping.
- Introduction part is very conceptual and informative. However, this review has some limitations. The authors may add future prospective and challenges.
>>>The last section is now labeled as “6. Conclusions and future prospects”
The section uses Box 1 to present the benefits and challenges of genomics in drug development. The text now also includes the following assessment of the future.
“Genomics (omics) technologies are becoming an integral component of drug development. They respond to the goal of compressing drug development timelines, and reflect the attention to personalized care.”
- Check syntax errors and misspells in whole manuscript.
>>>Done